# Pituitary Adenoma in the Philippines: A Scoping Review on the Treatment Gaps, Challenges, and Current State of Care

**DOI:** 10.3390/medsci12010016

**Published:** 2024-03-19

**Authors:** Mykha Marie B. Tabuzo, Mary Angeline Luz U. Hernandez, Annabell E. Chua, Patricia D. Maningat, Harold Henrison C. Chiu, Roland Dominic G. Jamora

**Affiliations:** 1Division of Adult Neurology, Department of Neurosciences, College of Medicine and Philippine General Hospital, University of the Philippines Manila, Manila 1000, Philippines; mbtabuzo@up.edu.ph; 2Division of Neurosurgery, Department of Neurosciences, College of Medicine and Philippine General Hospital, University of the Philippines Manila, Manila 1000, Philippines; muhernandez@up.edu.ph (M.A.L.U.H.); aechua@up.edu.ph (A.E.C.); 3Division of Endocrinology, Diabetes, and Metabolism, Department of Medicine, College of Medicine and Philippine General Hospital, University of the Philippines Manila, Manila 1000, Philippines; tmaningat@gmail.com (P.D.M.); hcchiu@up.edu.ph (H.H.C.C.); 4Center for Diabetes, Thyroid and Endocrine Disorders, St. Luke’s Medical Center—Global City, Taguig 1634, Philippines; 5Department of Internal Medicine, Cardinal Santos Medical Center, San Juan City 1502, Philippines

**Keywords:** pituitary adenoma, treatment gaps, healthcare

## Abstract

Background: Pituitary adenomas are benign brain tumors that impose a heavy burden on patients worldwide. The local burden of disease is yet to be established due to scarcity of data. In line with this, this study aims to present the challenges and gaps in the treatment of pituitary adenomas in the Philippines. Methods: A scoping review of available relevant literature on epidemiology, clinical experience with treatment, health financing, and healthcare delivery system based on the Preferred Reporting Items for Systematic reviews and Meta-analysis guidelines extension for Scoping Reviews was conducted. Results: The scarcity of updated local clinical data, inequity of distribution of resources, inadequate government support, and lack of affordable diagnostic testing, medications, and neurosurgical procedures are the factors that hinder provision of adequate care of pituitary adenomas in the Philippines. Conclusion: There are notable treatment gaps in the management of pituitary adenomas in the Philippines, which may be addressed by strengthening universal healthcare. Strategies to address these gaps were proposed, including improving public-private insurance coverage, increasing manpower, enhancing accessibility to resources, and spreading more awareness.

## 1. Introduction

Pituitary adenomas are primary brain tumors that are usually benign. With its increasing incidence and prevalence, and its potential to cause significant morbidity, this type of tumor may impose a heavy burden on both patients and their families as well as on health care systems [1]. Symptoms usually arise from compression of adjacent structures (blurring of vision, visual field defects, diplopia, increased intracranial pressure) as well as signs and symptoms of deficient or excess hormone production [2,3,4]. In recent studies, pituitary adenomas have been found to comprise approximately 10–15% of brain tumors, being the third most common intracranial neoplasm after meningiomas and gliomas [2,5]. With the widespread use of better diagnostic modalities and imaging techniques, prevalence has also increased [2]. Locally, there is still a paucity of epidemiological data for pituitary adenomas.

Treatment options for pituitary adenomas include surgical excision, medical therapies, and radiotherapy. In general, surgery is recommended primarily in symptomatic patients with nonfunctioning pituitary adenomas, acromegaly, and Cushing’s disease [3,6]. It is, however, also offered for prolactinomas when they have acute visual impairment, resistance or drug intolerance to medical treatment with dopamine agonists (DA), and when patients are predicted to achieve a good surgical outcome and do not want to pursue systemic therapy [2,3,7].

The burden of disease is still considered severe and chronic as these patients usually need life-long care and treatment; hence, it is important to determine and bridge the gaps in the management of pituitary adenoma [8]. To date, there is no report available on the treatment gaps in the management of pituitary adenomas in the Philippines. Therefore, this study aimed to review the present health care service delivery for patients with pituitary adenomas in the Philippines, and to identify any existing gaps and challenges in the management of these tumors.

## 2. Methods

### 2.1. Protocol

The authors used the Preferred Reporting Items for Systematic reviews and Meta-Analyses extension for Scoping Reviews (PRISMA-ScR) guidelines in performing this review [9]. The authors confirm that this scoping review was prepared in accordance with the Committee on Publication Ethics rules and regulations. Given the nature of this study, institutional and ethics review board approval was not required.

### 2.2. Eligibility Criteria

We included systematic reviews, meta-analyses, guidelines, randomized controlled trials, retrospective cohort studies, case series and reports, and textbooks written in the Philippines and by authors affiliated with institutions in the country. Studies involving adult and pediatric subjects were included. Studies using animal subjects were also included. There was no limitation on the publication year of the studies. We excluded abstracts and articles that were not written in English.

### 2.3. Information/Data Sources

International medical databases including Pubmed, SCOPUS, EBSCO, ClinicalTrial.gov, Western Pacific Region Index Medicus, and EMBASE were used in searching for relevant studies. The local medical database, the Health Research and Development Information Network, was also used to find studies. Available and relevant literature via official websites, including the Philippine Department of Health (DOH), Philippine Health Insurance Corporation (Philhealth), Philippine Statistics Authority (PSA), and medical associations (Philippine Neurological Association, Academy of Filipino Neurosurgeons, and Philippine College of Endocrinology, Diabetes, and Metabolism), was obtained.

### 2.4. Search and Selection of Sources

A scoping review of literature from the earliest indexed record of the databases until 31 August 2023 was conducted. The search terms (‘pituitary adenoma’ OR ‘pituitary macroadenoma’ OR ‘pituitary microadenoma’) AND ‘Philippines’ were used. Database searches were combined. Titles and abstracts were evaluated and were screened based on the eligibility criteria. After duplicates were excluded, full-text articles were reviewed.

### 2.5. Charting of Data and Synthesis of Results

Pertinent information regarding legislation, epidemiology, healthcare financing, information systems, workforce, pharmacotherapy, surgical management, and healthcare services in the local treatment of pituitary adenoma was obtained. Authors, titles, and institutional affiliation were extracted from the published studies. Data on the availability and cost of diagnostics, medications, and surgeries were also acquired from private and government medical institutions. The core topics were analyzed using a qualitative content approach. Data obtained were narratively described and summarized into figures and tables. A conceptual framework was produced to show the determined treatment gaps and challenges.

## 3. Results

### 3.1. Search of Studies

A total of 63 articles were obtained from the search (Figure 1), with 49 articles remaining after duplicates were removed. Thirty articles not related to pituitary adenomas in the Philippines were excluded. Nineteen articles were assessed for eligibility, with two articles excluded because they had no Filipino authors and four articles had only abstracts available. A total of 13 articles were included for synthesis.

### 3.2. Epidemiology of Pituitary Adenoma in the Philippines

Currently, there is no available epidemiological study of pituitary adenoma in the Philippines, and only hospital-based reports and data are accessible. A 2003 study looked at 120 patients diagnosed with pituitary adenomas at the Philippine General Hospital (PGH). Sixty-two percent of the patients were female, while thirty-eight percent were male, with ages ranging from 15 to 75 years old [2]. The most common presenting symptoms were headache (*n* = 65/120, 54%) and visual disturbances (*n* = 50/120, 42%). Of the 120 patients, 15% presented with acromegaly with elevated growth hormones and only two had cushingoid features. Cranial imaging findings showed that 90% were macroadenomas, with 94% suprasellar at the time of diagnosis. Eighty-four patients underwent surgery, with eighty-two percent (*n* = 69/84) of the procedures performed via a transsphenoidal approach [2]. Another retrospective study conducted in a private tertiary institution with 45 patients showed that males and females were almost equally affected, with ages ranging from 31 to 60. The most common symptoms were visual disturbances (*n* = 16/45, 35%) and headache (*n* = 13/45, 28%). Cranial imaging findings were consistent with a pituitary adenoma in 73% (*n* = 33/45) of patients, with the majority having microadenomas (*n* = 24/33, 73%) [10]. Seventy-six percent of the patients (*n* = 34/45) underwent surgery, mostly through the transsphenoidal approach (*n* = 22/45, 49%). Eleven percent (*n* = 4/45) had subsequent gamma knife stereotactic radiation [10].

In a 2013 case series conducted in PGH on patients presenting with Cushing’s syndrome, adrenocorticotropic hormone (ACTH)-producing pituitary adenomas were found to be the most common cause of Cushing’s syndrome, comprising 42% (*n* = 8/19) of cases [11]. This is consistent with the usual etiologies of endogenous Cushing’s syndrome worldwide.

### 3.3. Other Local Researches in Pituitary Adenoma

As of August 2023, there were only thirteen indexed research studies from the Philippines: five case reports (Table 1), six retrospective studies, one systematic review, and one cross-sectional study (Table 2). Studies showed varied approaches on medical and surgical management of different types of pituitary adenoma. Despite the limited resources, complicated cases such as giant adenomas, pituitary apoplexy, and acute corticotropic insufficiency had good outcomes [11]. Table 1 shows the summary of the clinical profile and the outcomes of patients with pituitary adenoma in the Philippines from published case reports.

### 3.4. The Healthcare System of the Philippines

Healthcare in the Philippines is delivered through a dual system composed of the public sector and the private sector [19]. The DOH supervises the government corporate hospitals as well as specialty and regional hospitals, while local government units (LGUs) manage the district and provincial hospitals and other rural health units (Figure 2) [19]. This system is dependent on the premise that the LGUs are the most informed of their constituents’ health needs, thus being able to immediately respond and provide for their needs [19,20]. In addition, the topography of the Philippines, being an archipelago with 7641 islands, also contributes to the maldistribution of health facilities, health personnel, and specialists, especially in isolated rural areas [19].

### 3.5. Healthcare Financing and Coverage in the Philippines

Philippine healthcare financing is a combination of the Beveridgean system (government tax-funded financing of DOH and LGU health facilities), the Bismarckian system (PhilHealth premium- and tax-funded financing), small-pooled private payment schemes, and large unpooled financing, which consists of out-of-pocket expenses as shown in Figure 3 [12].

In 2021, the Current Health Expenditure (CHE) of the Philippines reached USD 19.2 billion (USD 1.00 = PHP 56.60 as of 26 August 2023), which was 18.5% higher than in 2020. On average, every Filipino spent USD 173.80 for healthcare [21]. In the past decade, the government’s contribution to healthcare financing has increased from 21% to 38.7%, while out-of-pocket spending has decreased from 58.9% to 45.0% [22]. Although government health expenditure has increased significantly, it remains outstripped by private sector funding sources [19]. PhilHealth introduced social health insurance in 1995 to serve as coverage for Filipinos not able to financially cover their healthcare needs. This government health insurance system reportedly provides coverage for 92% of the population, of which 40% belong to the poor sector of the population, fully subsidized by the government [19]. However, services that are covered are mostly inpatient care, with outpatient care costs provided only for lower income members [19]. Even with the expansion of Philhealth coverage, out-of-pocket payments remain high (44%), serving as the major source of funding for healthcare [19,20].

PhilHealth coverage for central nervous system neoplasms are divided into benign or malignant cases. Pituitary tumors fall under the category of benign neoplasms of endocrine glands with a case rate of USD 211.90. Complications arising from pituitary adenoma treatment that are covered by PhilHealth are under disorders of the pituitary gland, with a case rate of USD 208.40, or Cushing’s syndrome, with a case rate of USD 167.80 [23].

In 2019, the “Malasakit” (Concern) Centers were established in all DOH hospitals in the country as well as in PGH as mandated by Republic Act No. 11463. They serve as a one-stop shop for medical and financial assistance from the DOH, Department of Social Welfare and Development, Philippine Charity Sweepstakes Office, and PhilHealth. They will also facilitate access to financial assistance programs provided by other government agencies, LGUs, nongovernmental organizations, and private institutions and individuals [24].

There are also non-government agencies that provide funding for healthcare of patients with neurologic diseases, such as the Philippine Brain Tumor Alliance, Sagip Buhay Medical Foundation Inc. (Manila, Philippines), and Let’s Save the Brain Foundation; these help in funding diagnostic workups, treatment regimens, and surgical needs of patients [20,25,26,27].

### 3.6. Specialist Training and Medical Education

The diagnosis of pituitary adenomas requires a multidisciplinary approach involving a group of specialists, including endocrinologists, neurologists, neurosurgeons, radiologists, radiation oncologists, and neuropathologists, working together to develop an “individualized patient-centric” approach [28,29]. In the Philippines, patients usually seek consult for neurologic symptoms such as headache, blurring of vision, and other symptoms caused by mass effect, so they are initially seen by neurologists and ophthalmologists. With history and neurologic physical examination, neurologists can localize the lesion and advise the appropriate imaging. Patients are referred to endocrinologists for comprehensive evaluation of pituitary function and treatment. Most patients are seen in close collaboration with neurosurgeons for possible surgical options. They are also referred to other specialty services like neuroradiology radiation oncology. Once the tumor is excised, neuropathologists will be able to confirm the diagnosis.

Currently, the ratio of medical doctors per 10,000 patients in the Philippines is 7.86 [30]. As of October 2023, there are 702 PNA board-certified adult neurologists and 100 board-certified pediatric neurologists, but the distribution is skewed towards the urban areas, with approximately 45% working in the National Capital Region (NCR) [31,32,33]. One adult neurologist is available for every 140,000 adult Filipinos, while one pediatric neurologist for every 315,000 Filipino children.

As of October 2023, there are 174 board-certified neurosurgeons practicing in the Philippines, or approximately one neurosurgeon for every 600,000 Filipinos [34]. There are eleven accredited training institutions for adult neurology, each with four to five graduates per year; eleven institutions for neurosurgery, each with one to three graduates per year, and three institutions for pediatric neurology in the Philippines, each with one to three graduates per year [33,34]. Currently, there are only 525 endocrinologists in the Philippines, with 422 who are board-certified mostly practicing in the NCR [35]. In the country, there are now 12 training institutions under the Philippine College of Endocrinology, Diabetes and Metabolism, each with an average of 2 to 6 graduates per institution per year, a total of 25–30 graduates per year [35]. On the other hand, there are 112 practicing radiation oncologists distributed in 51 facilities throughout the country at present [14,36]. There are nine residency training programs that produce an average of one to two graduates per year [36]. In terms of diagnostic medicine, there are around 500 registered members of the Philippine Society of Pathologists, three of whom have fellowships in neuropathology [37]. There are 32 accredited residency programs for pathology in the country, but there are no available local fellowships in neuropathology [37]. At present, there are a total of 2025 members of the Philippine College of Radiology [12]. Table 3 summarizes the number of specialists included in the management of pituitary adenoma compared to the Filipino population.

### 3.7. Challenges on Diagnostics

ACTH-producing pituitary adenoma is the most frequent cause of Cushing’s Syndrome [11,38]. Following the algorithm for assessment of patients suspected with Cushing’s Syndrome, 24-hr urinary free cortisol, overnight dexamethasone suppression test, and serum ACTH are necessary diagnostic tests [11,13]. Midnight salivary cortisol may also be used; however, it is available only in selected laboratories [13]. Evaluation of levels of different hormones such as prolactin, thyroid-stimulating hormone (TSH), free T4, follicle-stimulating hormone (FSH), estradiol or testosterone, ACTH, growth hormone (GH), insulin growth factor-1 (IGF-1), and fasting early morning cortisol are also necessary [29]. Cranial magnetic resonance imaging with contrast is the primary imaging tool to document the presence of a pituitary lesion, and the presence of any hemorrhage [26,39]. Moreover, state of the art high-resolution MRI of the pituitary gland at 3 Tesla (3T) is now available, which produces higher resolution images of the pituitary gland that allow evaluation of subtle differences between normal and abnormal tissue [40]. Simultaneous bilateral inferior petrosal sinus sampling (BIPSS) is another crucial diagnostic work-up of Cushing’s syndrome since it is the most accurate procedure in the evaluation of hypercortisolism of pituitary or ectopic origin, relative to clinical, biochemical, and imaging analyses, with a sensitivity and specificity of 88–100% and 67–100%, respectively [41]; however, there is no available report of it being conducted in the Philippines.

In the Philippines, endocrine tests are offered in laboratories and large tertiary centers that are usually privately-owned, with prices ranging from USD 6 to 57 [40]. In 2016, DOH reported that there were 429 CT machines (4.2 machines per 1 million population) and 78 magnetic resonance imaging (MRI) units (0.8 MRI units per 1 million population) across government hospitals as well as private imaging facilities [42]. Price ranges from USD 150 to 200, depending on the institution [30]. Additionally, advanced imaging technology such as the 3T MRI is only available in one private tertiary hospital in the country and is not readily accessible to the average Filipino [43]. Table 4 shows the summary of estimated prices of some diagnostic tests needed for the evaluation of pituitary adenomas. Based on current medical case rates, neurosurgical procedures, specifically craniotomy and biopsy with or without excision of tumor, have rates ranging from USD 2060 to 4120 [23,31].

### 3.8. Challenges on Treatment

A multidisciplinary approach is necessary for the ideal treatment of pituitary adenomas. For non-functioning macroadenomas, prompt surgical resection via a transsphenoidal approach or craniotomy is recommended in patients with visual abnormalities such as visual field defect and ophthalmoplegia [16,17], and more emergently in patients who present with pituitary apoplexy, loss of endocrine function, and significant tumor growth [15,18,28,44,45]. Locally, the transsphenoidal technique is the most common, which is consistent with the world standard since it can be used effectively for 95% of pituitary tumors [15,16,17,18,44,45,46,47]. Transcranial and endoscopic techniques are also used in more appropriate circumstances [15,16,17,18,44,45,46]. Studies showed that pituitary surgery in the Philippines is relatively a safe procedure with the highest reported mortality rate of 3% and morbidity rate of 30% [1,16,17]. Local mortality and morbidity rates are higher compared to the global overall mortality rate for transsphenoidal surgery, which is less than 0.5% [46]. A retrospective study conducted in another third world country revealed a mortality rate of 1.6% and morbidity rate of 6.5% among patients who underwent pituitary surgery using transcranial, transsphenoidal, and endoscopic techniques [47]. The PhilHealth coverage for excision of pituitary tumor via a transsphenoidal approach only amounts to USD 900, which at most covers less than half of the minimum cost of surgery [23].

For patients with functioning tumors, management depends on the hormone involved. For prolactinoma and acromegaly, DAs are used as first-line treatment [15,28]. Currently available DAs are cabergoline and bromocriptine. Cabergoline is more than 90% effective in normalizing prolactin levels and decreasing tumor size, with the advantage of being given only twice weekly; unfortunately, it is expensive, not readily available in the country, and must be imported from other countries like India [28,45]. Bromocriptine is available in some local drugstores, costing around USD 2 per tablet [46]. For patients with persistently elevated IGF-1 and GH, somatostatin analogs are given, such as octreotide, lanreotide, and pasireotide. Short-acting and long-acting octreotide formulations are available in the Philippines, with prices ranging from USD 10 to 18 per 0.1 mg/mL 1 mL ampule [48]. It can be subsidized occasionally through medical social services but is not sustainable on a long-term basis as the cost remains prohibitive. Lanreotide autogel 90 mg pre-filled formulation was previously available in the country. However, its FDA approval expired in 2021 [49]. For patients with recurrent and refractory Cushing’s disease, drug therapy that targets steroidogenesis such as ketoconazole and metyrapone may be used. However, there is a low utilization rate in our setting on top of the difficulty in access and high costs. For patients with non-functioning tumors presenting with hormonal deficiencies, lifelong monitoring of hormone levels is necessary to ensure appropriate and correct dosing of hormone replacement therapy. Regular monitoring of hormones entails additional costs on top of large out-of-pocket expenses for hormone replacement. The most common hormone deficiencies associated with large non-functioning adenomas include cortisol, thyroid hormones, and sex steroids. Glucocorticoid replacement using hydrocortisone, prednisone, and prednisolone are often the agents of choice. Both prednisone and prednisolone are available in most pharmacies, but hydrocortisone must be purchased from special compounding pharmacies. All these agents are relatively affordable. Thyroid hormone replacement in the form of levothyroxine is accessible and affordable for most patients. Sex hormone replacement utilizes testosterone injections (testosterone enanthate, cypionate, or undecanoate) for men and oral (estradiol valerate) or injected estradiol (estradiol benzoate or equivalent ester) for premenopausal women. Unfortunately, most of these sex hormone replacement therapies are not readily available in the local setting. Patients in need of these replacement regimens often purchase from online sellers and bodybuilding or supplement shops.

Radiotherapy (RT) is considered in patients with recurrent tumors and progressive residual tumors [27]. As of 2019, there are 50 RT-capable centers in the Philippines. However, most are located in the Metro Manila area, with 1–2 major hospitals in the other two major islands in the country having this capability (Cebu in the Visayas and Davao in Mindanao) [30]. The cost for RT depends on the number of sessions, usually costing USD 2000–4000 for the entire treatment duration [30,44]. In addition, patients undergoing radiotherapy should subsequently be monitored for the development of panhypopituitarism, which occurs approximately 5–10 years post-RT. Replacement of deficient hormones with appropriate monitoring is needed for this subset of patients.

## 4. Discussion

Pituitary adenomas require collaborative efforts of specialists with a combined surgical and medical approach. The necessary diagnostics and treatment modalities are available in the Philippines, but access remains limited to urbanized centers as in other low to middle income countries, thus adding to the difficulty in decreasing the burden of the disease. Updated research on pituitary adenomas and their management is also sparse. There are no high-volume specialized pituitary centers where such data can be generated. Despite policies being in place, timely diagnosis and universal access to treatment for patients with pituitary adenomas remain difficult.

Tumor excision and medical control of clinical symptoms are the mainstays in the treatment of pituitary adenomas. These treatment modalities are expensive in the Philippines due to limited insurance coverage from government or private sources, and expenses are shouldered mostly by the patients themselves. Currently, there are no studies on the comparative cost of treatment in Southeast Asia. Access to and cost of medications also pose a challenge, with most first-line drugs unavailable locally. Logistically, most of the brain surgery and radiotherapy capable secondary and tertiary medical centers are concentrated in urban centers, thus rendering treatment options even less accessible to most patients belonging to the lower socioeconomic strata. The scarcity of specialists also has a significant impact since a collaborative approach is necessary for adequate management of the disease. There are no data yet for the ideal ratio of specialists to patients; however, according to the World Health Organization, the ideal doctor-patient ratio is 10:10,000 [29]. In 2021, the Philippines had 7.86 medical doctors per 10,000 population [29].

In general, treatment of diseases in the Philippines faces economic, administrative, and social challenges. Figure 4 shows the summary of proposed challenges in the current state of healthcare for patients with pituitary adenomas in the Philippines. Some recommendations on how to address these gaps in treatment are also included. Our review and qualitative analysis revealed that health financing and manpower are the major areas for improvement in the effective management of patients with pituitary adenoma. Access to hospitals where treatments are available, including surgery and radiotherapy, is a resolvable issue. This can be addressed by increasing government healthcare insurance coverage and urging private insurance companies to cover non-emergency cases for brain surgery [30]. In addition, establishment of larger centers across the country to decongest the already crowded and long queues in Metro Manila. The unavailability of some medications in the local market can be solved by increasing availability and volume of these drugs, especially in public pharmacies. Government academic centers together with the DOH can spearhead the procurement process based on need and utilization measures. Also, establishing more training programs to produce more pathology, radiology, radiation oncology, endocrinology, neurology, and neurosurgery graduates may help address the lack of specialists. Promoting international fellowships and training may help in attaining a global standard of care. Producing an updated database on epidemiology and local experience in managing pituitary adenomas will help formulate administrative programs that can further bridge gaps in treatment. Lastly, setting up a streamlined referral system or network based on the working database or registry across different centers that cater to the care of patients with pituitary adenoma will help improve our process flow.

One limitation of this review is the unavailability of access to unpublished studies. Despite our efforts to be as comprehensive as possible, this review may not have retrieved all pertinent studies in the published, unpublished, and unreported literature. The search algorithm and the keywords that were used may have failed to include other pertinent terms. Only articles published in English were included, thus some studies written in other languages may have been missed. Moreover, the reviewer used her judgment to determine whether the studies have met the criteria and may be subject to reviewer bias.

To our best knowledge, this is the first scoping review that described gaps and challenges in the treatment of pituitary adenomas in the Philippines. Findings of this study advocate better recognition for patients, clinicians, researchers, and legislators alike.

## 5. Conclusions

There are gaps in the management of pituitary adenomas in the Philippines. These include availability and affordability of medications, lack of manpower, as well as inequity in access to facilities, inadequate coverage of government and private healthcare insurance, and scarcity of local studies and epidemiologic data.

## Figures and Tables

**Figure 1 medsci-12-00016-f001:**
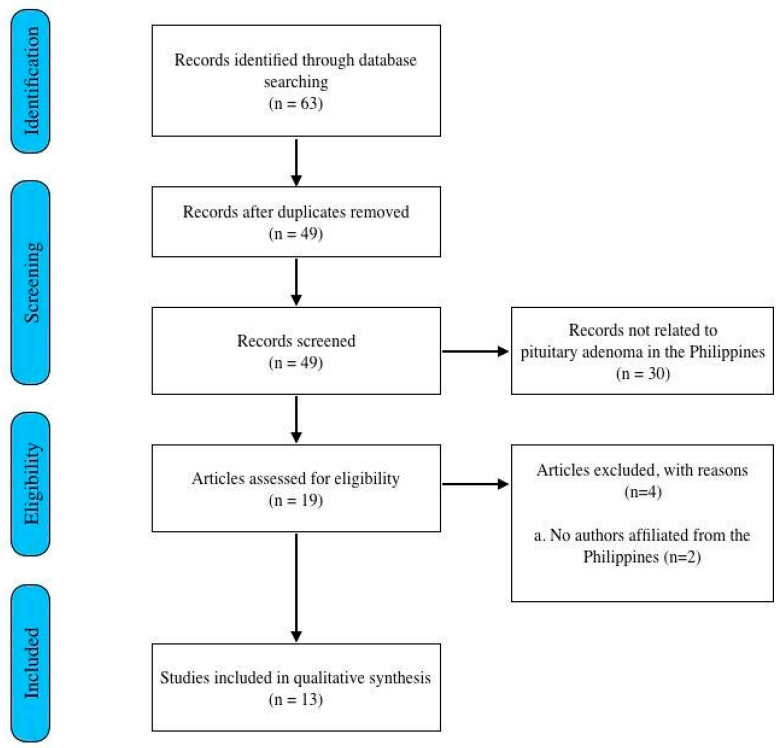
Flow chart of methodology adapted from the PRISMA guidelines for scoping reviews.

**Figure 2 medsci-12-00016-f002:**
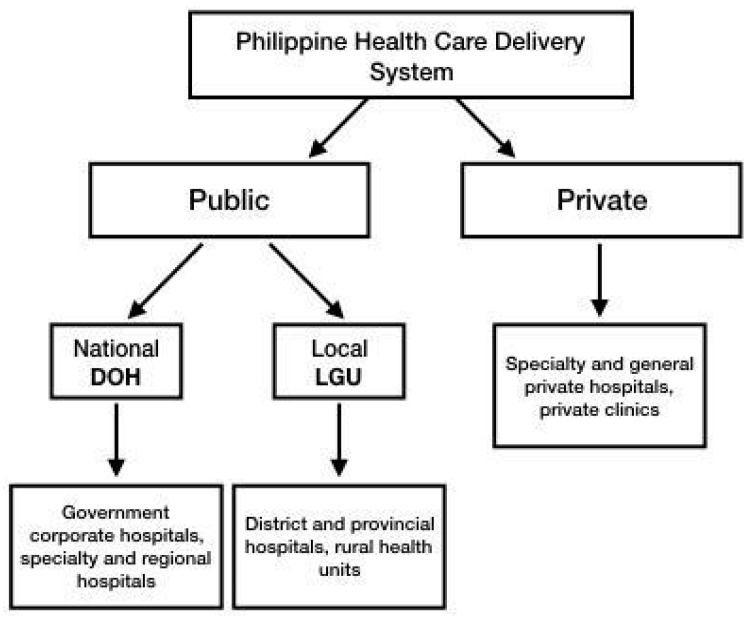
Diagram summarizing the health care delivery system in the Philippines.

**Figure 3 medsci-12-00016-f003:**
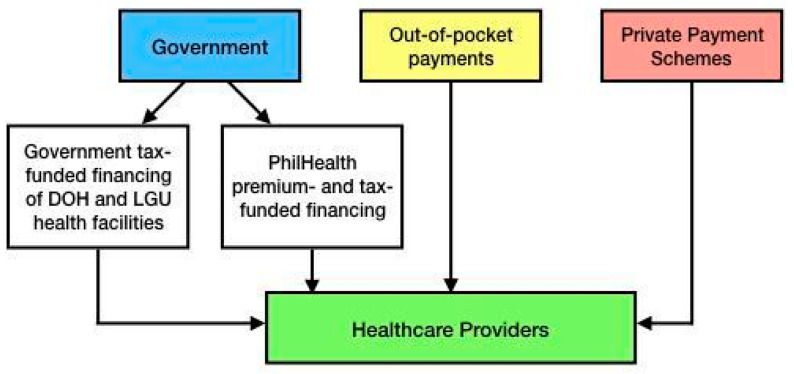
Diagram showing the outline of the health financing system of the Philippines.

**Figure 4 medsci-12-00016-f004:**
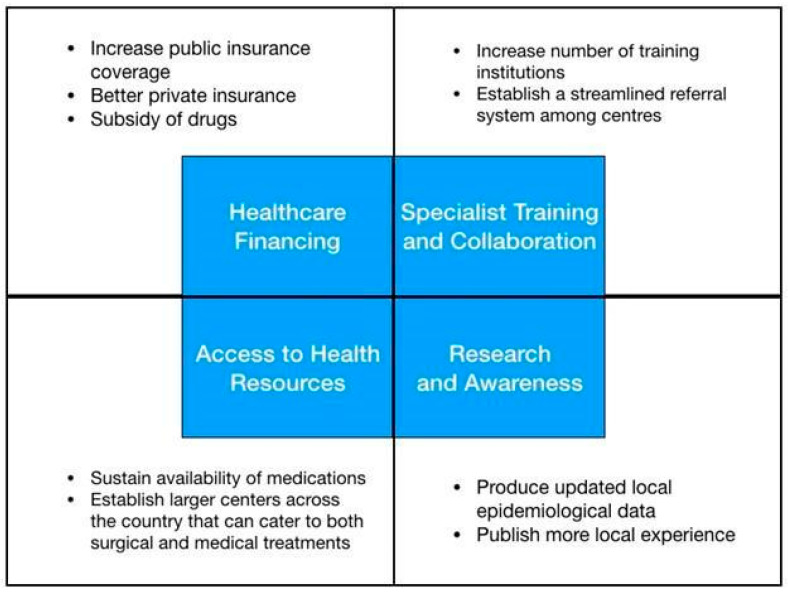
Conceptual framework exhibiting gaps in the treatment of pituitary adenoma in the Philippines.

**Table 1 medsci-12-00016-t001:** Summary of clinical outcomes of pituitary adenoma cases.

Author	Age/ Sex	Type of Adenoma	Clinical Presentation	Imaging Findings	Hormonal Rofile	Surgical Treatment	Medical Treatment	Outcome
Bunoy et al., 2012 [4]	38/M	macro- adenoma	seizures, frontal headache, blurring of vision (bitemporal hemianopsia)	4.0 × 5.0 × 4.5 cm sellar-suprasellar mass	low cortisol, high TSH, normal ACTH	cranioto- my with excision of the tumor	methima- zole, prednisone	Improved
Tating et al., 2016 [12]	13/M	micro- adenoma	bilateral lower extremity weakness, hypertension	0.8 × 0.9 × 0.8 cm anterior pituitary gland mass	high ACTH, high cortisol	transphenoidal adenomectomy was indicated but deferred because dexamethasone suppression test revealed high serum cortisol and 24-h free urine cortisol	none	Expired
Sandoval et al., 2020 [13]	37/F	Macro- adenoma	amenorrhea, progressive weight gain, facial roundness, leg weakness, easy bruisability, blurring of vision (bitemporal hemianopsia), hyperpigmented fingernails	4.6 × 4.1 × 7 cm lobulated, heterogeneously enhancing sellar-suprasellar mass with cystic and necrotic components	High ACTH	transphenoidal excision	none	Improved
Mendo-za et al., 2015 [14]	46/F	macro- adenoma	amenorrhea, acromegaly	2.1 × 3.3 × 2.4 cm sellar-suprasellar mass with foci of intratumoral hemorrhages suggestive of subclinical pituitary apoplexy	high GH, high prolactin	none	cabergo- line 0.5 mg tablet once a week	Improved
Jordan et al., 2022 [15]	40/F	Giant adenoma	acromegaly, headache, blurring of vision (bitemporal hemianopsia)	6.4 × 7.0 × 5.5 cm lobulated pituitary mass with cystic degeneration and necrosis	high GH, high IGF-1	surgical resection via transcranial approach	bromocriptine, radiotherapy	Improved

**Table 2 medsci-12-00016-t002:** Summary of other indexed articles.

Author	Title	Journal	Institution	Publication Year	Study Design
Alinsonorin, et al. [2]	Pituitary adenoma: clinical profile of 120 patients at the Philippine General Hospital	Philippine Journal of Internal Medicine	Philippine General Hospital	2003	Retrospective, descriptive
Seng, et al. [3]	Extracapsular resection of pituitary adenomas: a systemic review	Asian Journal of Neurosurgery	Philippine General Hospital	2023	Systematic review
Faltado, et al. [6]	Factors associated with postoperative diabetes insipidus after pituitary surgery	International Journal of Endocrinology and Metabolism	Philippine General Hospital	2017	Retrospective cohort
Villegas, et al. [10]	A review of patients with pituitary tumors at St. Luke’s Medical Center from January 1997 to September 2000	Philippine Journal of Internal Medicine	St. Luke’s Medical Center	2002	Retrospective study
Lo, et al. [11]	Endogenous Cushing’s syndrome: the Philippine General Hospital experience	Journal of ASEAN Federation of Endocrine Societies	Philippine General Hospital	2014	Cross-sectional
Cudal, et al. [16]	Postoperative complications of trans-sphenoidal surgery in a local tertiary hospital during hospital stay	Philippine Journal of Internal Medicine	Makati Medical Center	2018	Retrospective cross- sectional
Carampatana-Jandug, et al. [17]	In-hospital postoperative complications in patients with pituitary adenoma who underwent pituitary surgery from January 2010 to December 2015: a multicenter study	International Journal of Endocrinology and Metabolism	Chong Hua Hospital	2017	Retrospective cohort
Fonte, et al. [18]	Treatment outcomes of pituitary tumors at the University of Santo Tomas Hospital: 2004–2008	Philippine Journal of Internal Medicine	University of Santo Tomas Hospital	2009	Retrospective cohort

**Table 3 medsci-12-00016-t003:** Ratio of specialist to the Filipino population.

Specialist	Specialist to Filipino Ratio
Endocrinologist	1:198,000
Adult Neurologist	1:140,000
Neurosurgeon	1:600,000
Pediatric Neurologist	1:315,000
Radiologist	1:51,555
Radiation Oncologist	1:900,000
Neuropathologist	1:34,800,000

**Table 4 medsci-12-00016-t004:** Price range of diagnostic tests.

Laboratory Test/Imaging	Cost Government-Subsidized (USD)	Cost No Government No Subsidy (USD)	Cost Private (USD)
IGF-1	NA	NA	175.00
ACTH	NA	NA	136.90
Cortisol	NA	NA	52.80
Prolactin	5.80	8.50	9.10–31.20
FSH	6.10	9.70	11.50–24.20
LH	9.20	12.40	13.00–24.20
Estradiol	6.90	10.60	11.10–46.80
Testosterone	15.30	32.20	33.80–46.80
TSH	6.10	10.50	12.10–14.40
FT3	8.30	12.90	13.60–49.90
FT4	6.40	10.00	10.50–49.90
Cranial MRI with contrast	153.70	179.30	190.00–210.90
Cranial MRI—plain	75.60	91.30	96.60–107.30
Cranial CT scan with contrast	86.30	98.10	103.00–112.80
Cranial CT scan—plain	26.30	54.40	65.30–73.50

CT—computerized tomography, MRI—magnetic resonance imaging, NA—not available, IGF-1—Insulin-like growth hormone-1, ACTH—Adrenocorticotropic hormone, FSH—follicle stimulating hormone, LH—luteinizing hormone, TSH—thyroid stimulating hormone, FT3—free triiodothyronine, FT4—free thyroxine hormone.

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
