# Peer review of "Pituitary Adenoma in the Philippines: A Scoping Review on the Treatment Gaps, Challenges, and Current State of Care"

_medsci, 2024, doi:10.3390/medsci12010016_

Round 1

Reviewer 1 Report

Comments and Suggestions for Authors

Tabuzo et al. present their study on pituitary adenoma treatment in the Philippines. Great topic, very interesting insights and I think the paper has many benefits and adds new evidence to the current literature. Meanwhile, I think a few comments could improve the manuscript:

- In the introduction: prolactinomas are not only treated in cases of drug intolerance or resistance. We now have evidence that they can also be treated surgically in cases of visual impairment or in cases in which patients do not want to pursue systemic therapy 

- In the results section you can remove the part "this section may be divided (...) that can be drawn

- In all your results, you write "n=XX, ..%". The percentages are always different, but the n number is the same. I think you aim to describe the number of patients included in the analysis, but please refer to (n=72/120, 54%) for example.

- What is the difference between Table 1 and 2? What are you trying to show?

- Sections 3.4 - 3. 6 are very interesting and deserve more visualisation of the data. Maybe a Table listing the most important facts?

- You have a section 3.6 twice.

Your work has merit; I think it was very fascinating to be informed about the challenges in the Philippines, especially from a high-income country point of view. Great idea!

Author Response

Comments

Response/Revision

In the introduction: prolactinomas are not only treated in cases of drug intolerance or resistance. We now have evidence that they can also be treated surgically in cases of visual impairment or in cases in which patients do not want to pursue systemic therapy

Thank you for your comments.

This has been revised to: “In general, surgery is recommended primarily in symptomatic patients with nonfunctioning pituitary adenomas, acromegaly, and Cushing’s disease [3,6]. It is, however, offered for prolactinomas when they have acute visual impairment, resistance or drug intolerance to medical treatment with dopamine agonists (DA), and when patients are predicted to achieve a good surgical outcome and do not want to pursue systemic therapy.”  (p. 1, Line 103-108)

In the results section you can remove the part "this section may be divided (...) that can be drawn

We apologize for this mistake.

This part of the results section was already deleted.

In all your results, you write "n=XX, ..%". The percentages are always different, but the n number is the same. I think you aim to describe the number of patients included in the analysis, but please refer to (n=72/120, 54%) for example.

Thank you for your comments and suggestions.

The number of patients included in the analysis was revised to this format, for example: (n = 22/45, 49%). (p. 7, Line 180-192)

What is the difference between Table 1 and 2? What are you trying to show?

Thank you for your comment and clarification.

Table 1 shows the clinical profile, treatments done, and outcomes of the reported cases of pituitary adenoma in the Philippines while table 2 only enumerates the other different indexed studies about pituitary adenoma in the Philippines. To avoid confusion, case reports were excluded from table 2. (p. 9-10, Line 210-213)

Sections 3.4 - 3. 6 are very interesting and deserve more visualization of the data. Maybe a Table listing the most important facts?

Thank you for your comments and suggestions.

For the following sections, figures and tables were added for more visualization of the data.

3.4 Healthcare system in the Philippines

-included a diagram summarizing the healthcare delivery system in the Philippines (p. 11, Line 224-225)

3.5 Healthcare financing and coverage in the Philippines

-included a diagram showing the outline of the health financing system of the Philippines  (p. 11, Line 228-232, 269-271)

3.6 Specialist training and medical education

-included a table showing the ratio of number of specialists to the Filipino population (p. 14-15, Line 303-304)

You have a section 3.6 twice.

We apologize for this mistake.

The duplicated section was already deleted.

Reviewer 2 Report

Comments and Suggestions for Authors

The authors reviewed the challenges and gaps in the management of pituitary adenomas in the Philippines and described suggestions to improve the situation in the Philippines, which will lead to better care for patients.

As there are several flaws in this submitted manuscript, I could not recommend this in the publication in Medical Sciences.

Concept: This study reviews the challenges and gaps in the management of pituitary adenomas only in the Philippines, which seems vague. It would be better to compare the challenges and gaps in the Philippines with the ones in other countries. Also, are there any differences between the challenges and gaps in the management of pituitary tumors and other endocrine disorders? This study summarizes publications regarding epidemiology, case reports, retrospective studies, reviews, and a cross-sectional study and the healthcare system of the Philippines as well as medical education. It would describe the situation in the Philippines but would lack the novelty in the management of pituitary tumors and/or training in this field. 

Introduction: 

Line 45: Acromegaly should be mentioned in addition to nonfunctioning pituitary adenomas and Cushing's disease. 

Methods: 

Search and selection of sources: I was wondering why the term, "pituitary neuroendocrine tumor", was not used in this study.

Results:

Line 111-112: The most common presenting symptoms were headache (n = 120, 54%) and visual disturbances (n = 120, 42%). n would be 65 and 55, respectively?

Line 179-181: Radiologists and neuropathologists are also involved in the care of patients with pituitary tumors.

Line 179-200: How many cases with pituitary tumors would trainees (endocrinology and neurosurgery) see during the clinical fellowships?

Line 182-187: The main specialists for patients with pituitary tumors would be endocrinologists, neurosurgeons, neuropathologists, and radiologists, rather than neurologists.  How are neurologists involved in the care of pituitary patients in the Philippines? The role of neurologists in the care of pituitary patients is vague in the manuscript. This might be different from the one in other countries and advantageous!

Line 201-220: It would be better to mention the availability of 3T MRI scan and IPSS, as they are often required to localize the source of excess ACTH. 

Line 241-248: Isn't short-acting octreotide available in the Philippines? Could dopamine agonists and/or pegvisomant be used for acromegaly in the Philipines?

Although Table 1 summarizes case reports, Table 2 also contains case reports, which is strange and confusing.  

Author Response

Line 45: Acromegaly should be mentioned in addition to nonfunctioning pituitary adenomas and Cushing's disease.

Thank you for your comments.

This has been revised to:
“In general, surgery is recommended primarily in symptomatic patients with nonfunctioning pituitary adenomas, acromegaly, and Cushing’s disease.”

(p. 3 lines 103-104)

Methods: Search and selection of sources: I was wondering why the term, "pituitary neuroendocrine tumor", was not used in this study.

Thank you for your comments.

Although the International Pituitary Pathology Club in 2016 formally proposed new terminology for Pituitary adenoma (PA) to be pituitary neuroendocrine tumor (PitNET), the fourth edition of the WHO classification of PA released in 2017 did not adopt this nomenclature. There are still debates on whether to adopt this nomenclature so the researchers maintained the term PA and did not use PitNET at this time.

Line 111-112: The most common presenting symptoms were headache (n = 120, 54%) and visual disturbances (n = 120, 42%). n would be 65 and 55, respectively?

Thank you for your comments and clarification.

This has been revised to:

“The most common presenting symptoms were headache (n = 65/120, 54%) and visual disturbances (n = 50/120, 42%).”

(p. 7 lines 179-180)

Line 179-181: Radiologists and neuropathologists are also involved in the care of patients with pituitary tumors.

Thank you for your comments and suggestions.

We have included radiologists and neuropathologists in the manuscript.

(p. 13 lines 274-276)

Line 179-200: How many cases with pituitary tumors would trainees (endocrinology and neurosurgery) see during the clinical fellowships?

Thank you for your comment.

Unfortunately, there is currently no published data regarding this.

Line 182-187: The main specialists for patients with pituitary tumors would be endocrinologists, neurosurgeons, neuropathologists, and radiologists, rather than neurologists.  How are neurologists involved in the care of pituitary patients in the Philippines? The role of neurologists in the care of pituitary patients is vague in the manuscript. This might be different from the one in other countries and advantageous!

Thank you for your comments.

Patients with pituitary adenoma often see neurologists when they feel neurological manifestations such as headache, visual defect,

trigeminal sensory loss, and hydrocephalus. Neurologists then help in localization, diagnosis, and workup for these patients.

Line 201-220: It would be better to mention the availability of 3T MRI scan and IPSS, as they are often required to localize the source of excess ACTH.

Thank you for your comments and suggestions.

We agree, hence, have included the available data regarding the 3T MRI scan and IPSS.

This has been added:

“Moreover, state of the art high-resolution MRI of the pituitary gland at 3 Tesla (3T) is now available, which produces higher resolution images of the pituitary gland that allow evaluation of subtle differences between normal and abnormal tissue [36].  Simultaneous bilateral inferior petrosal sinus sampling (BIPSS) is another crucial diagnostic work-up of Cushing’s syndrome since it is the most accurate procedure in the evaluation of hypercortisolism of pituitary or ectopic origin, relative to clinical, biochemical and imaging analyses, with a sensitivity and specificity of 88–100% and 67–100%, respectively [37]; however, there is no available report of it done in the Philippines. “

(po. 15-16 lines 316-324)

“Additionally, advanced imaging technology such as the 3T MRI is only available in one private tertiary hospital in the country and is not readily accessible to the average Filipino [38].”

(po. 16 lines 330-332)

Line 241-248: Isn't short-acting octreotide available in the Philippines? Could dopamine agonists and/or pegvisomant be used for acromegaly in the Philippines?

Thank you for your comments and clarification.

On re-checking, there is indeed available short-acting octreotide in the Philippines. We apologize for that mistake. This has been revised accordingly in the manuscript.

“ Short-acting and long-acting octreotide formulations are available in the Philippines, with prices ranging from USD 10-18 per 0.1 mg/mL 1mL ampule”

(p. 18, line 371-373)

For acromegaly, DAs are also used as treatment [p. 18, line 364]; however, currently, there is no available published data regarding use of pegvisomant in the Philippines. This has been included in the manuscript as well.

Although Table 1 summarizes case reports, Table 2 also contains case reports, which is strange and confusing. 

Thank you for your comments.

Table 1 shows the clinical profile, treatments done, and outcomes of the reported cases of pituitary adenoma in the Philippines while table 2 only enumerates the other different indexed studies about pituitary adenoma in the Philippines. To avoid confusion, case reports were excluded from table 2. (p. 9-10, Line 210-213)

Reviewer 3 Report

Comments and Suggestions for Authors

The present PRISMA review about the current status of pituitary adenoma management in the Philippines is well written and documented; the authors made a nice and realistic picture of such disease management that give also an idea about the global status and advancement of neurosurgery in their country and I believe that they should be commended for their great job; on the other hand few minor problems are encountered as follow:

1) all manuscript section are too long and sometimes redundant please shorten as much as possible.

2) The only main feature of pituitary adenomas management that is practically missing and/or not very well described is the surgical technique used in Philippines countries, it is actually not clear if patient surgical management is in line with world wide standard of care (i.e. transcranial, transphenoidal, endoscopic approaches) as well as what is the patients outcome (please add data, figures and etc even from unpublished institutional data discussing their implications and if need for improvement by promoting international scholar/fellowship and/or other department cooperations).

3) Finally it is not clear if in this country are available or not medical/endocrinology high dependency unit to manage patients experiencing acute corticotropic insufficiency, pituitary apoplexy and/or giant pituitary adenoma (please add data, discuss, argument).

Author Response

Comments

Revision

All manuscript section are too long and sometimes redundant please shorten as much as possible

Thank you for your comments and suggestions.

Some sections have been revised to be less lengthy and redundant data were removed.

The only main feature of pituitary adenomas management that is practically missing and/or not very well described is the surgical technique used in Philippines countries, it is actually not clear if patient surgical management is in line with worldwide standard of care (i.e. transcranial, transsphenoidal, endoscopic approaches) as well as what is the patients outcome (please add data, figures and etc even from unpublished institutional data discussing their implications and if need for improvement by promoting international scholar/fellowship and/or other department corporations).

Thank you for your comments and suggestions.

We agree with your suggestion. We have discussed further the surgical technique used in the Philippines as well as the patient outcome in comparison to the global standard under the results section and discussion.

This has been added:
“Locally, the transsphenoidal technique is most commonly done, which is consistent with the world standard since it can be ​​used effectively for 95% of pituitary tumors [40-47]. Transcranial and endoscopic techniques are also used in more appropriate circumstances [40-46]. Studies showed that pituitary surgery in the Philippines is relatively a safe procedure with the highest reported mortality rate of 3% and morbidity rate of 30% [1,40,41]. Local mortality and morbidity rates are higher compared to the global overall mortality rate for transsphenoidal surgery which is less than 0.5% [46]. A retrospective study done in another third world country revealed a mortality rate of 1.6% and morbidity rate of 6.5% among patients who underwent pituitary surgery using transcranial, transsphenoidal, and endoscopic techniques [47].”

(p. 17, line 349-358)

Finally, it is not clear if in this country are available or not medical/endocrinology high dependency unit to manage patients experiencing acute corticotropic insufficiency, pituitary apoplexy and/or giant pituitary adenoma (please add data, discuss, argument).

Thank you for your comments and suggestions.

We have discussed further the surgical technique used in the Philippines as well as the patient outcome in comparison to the global standard under the results section and discussion.

This has been added: “cross-sectional study (Table 2). Studies showed varied approaches on medical and surgical management of different types of pituitary adenoma. Despite the limited resources, complicated cases such as giant adenomas, pituitary apoplexy, and acute corticotropic insufficiency had good outcomes [11]. “

(p. 7, line 203-206)

Round 2

Reviewer 2 Report

Comments and Suggestions for Authors

Thank you for the revised manuscript.

The authors revised most of the parts of the manuscript. However, I do not fully agree with the author's response about the neurologists' role.

Headaches and visual disturbance would lead to the arrangement of imaging studies. However, trigeminal sensory loss and hydrocephalus would rarely seen in patients with pitutiary tumor. The final diagnosis is made by neuropathologists. Also, in my understanding, workup including endocrine assessment is done by endocrinologists. The role of each specialist should be clearly documented. For instance,  neurologists screen patients from neurological findings and referring patients to endocrinologists, etc.

Author Response

Thank you very much for your comment. 

Management of pituitary adenoma is multidisciplinary. In the Philippines, patients usually seek consult for neurologic symptoms such as headache, blurring of vision, and other symptoms caused by mass effect so they are initially seen by neurologists and ophthalmologists. With history and neurologic physical examination, neurologists are able to localize the lesion and advise the appropriate imaging. Patients are referred to endocrinologists for comprehensive evaluation of pituitary function and treatment. Most patients are seen in close collaboration with neurosurgeons for possible surgical options. They are also referred to other specialty services like neuroradiology and radiation oncology. Once the tumor is excised, neuropathologists will be able to confirm the diagnosis.